Multi-omics analysis of pyroptosis regulation patterns and characterization of tumor microenvironment in patients with hepatocellular carcinoma

Shang Bingbing 1
Wang Ruohan 1
Qiao Haiyan 2
Zhao Xixi 1
Wang Liang 2 wangliang-dy@dmu.edu.cn
Sui Shaoguang 1 suishaoguang@163.com
1 Emergency Department, The Second Affiliated Hospital, Dalian Medical University , Dalian , China
2 Research and Teaching Department of Comparative Medicine, Dalian Medical University , Dalian , China
Date Swapneeta
Electronic publication date: 2023 May 11
Publication date: 2023
Volume: 11
Electronic Location ID: e15340
Received 2023 Jan 3; Accepted 2023 Apr 12
Copyright: © 2023 Shang et al.
Copyright year: 2023
Copyright holder: Shang et al.
License: This is an open access article distributed under the terms of the Creative Commons Attribution License, which permits unrestricted use, distribution, reproduction and adaptation in any medium and for any purpose provided that it is properly attributed. For attribution, the original author(s), title, publication source (PeerJ) and either DOI or URL of the article must be cited.
License URL: https://creativecommons.org/licenses/by/4.0/

Keywords: Hepatocellular carcinoma, Multi-omics analysis, Pyroptosis, Immune cells infiltration

Funding: The authors received no funding for this work.

==============================
Background

Hepatocellular carcinoma (HCC) is a primary malignant tumor of the liver, and pyroptosis has been identified as a novel cellular program that plays a role in numerous diseases including cancer. However, the functional role of pyroptosis in HCC remains unclear. The purpose of this study is to explore the relationship between the two found hub genes and provide targets for clinical treatment.

Methods

The Cancer Genome Atlas (TCGA) database was used to collect the gene data and clinically-related information of patients with HCC. After the differentially expressed genes (DEGs) were identified, they were intersected with the genes related to pyroptosis, and a risk prediction model was established to predict the overall survival (OS). Subsequently, drug sensitivity analysis, Gene Ontology (GO), Kyoto Encyclopedia of Genes and Genomes (KEGG), Gene Set Enrichment Analysis (GSEA), and Gene Set Variation Analysis (GSVA) was used to analyze the biological characteristics of the DEGs. Different immune cell infiltration and related pathways were analyzed, and hub genes were identified by protein-protein interaction (PPI). Finally, the expression of hub genes was verified by real-time quantitative PCR (qRT-PCR) and immunohistochemistry.

Results

We conducted a comprehensive bioinformatics analysis to investigate the molecular mechanisms of pyroptosis in hepatocellular carcinoma (HCC). A total of 8,958 differentially expressed genes were identified, and 37 differentially expressed genes were associated with pyroptosis through intersection. Moreover, we developed an OS model with excellent predictive ability and discovered the differences in biological function, drug sensitivity, and immune microenvironment between high-risk and low-risk groups. Through enrichment analysis, we found that the differentially expressed genes are related to various biological processes. Then, 10 hub genes were identified from protein-protein interaction networks. Finally, midkine (MDK) was screened from the 10 hub genes and further verified by PCR and immunohistochemistry, which revealed its high expression in HCC.

Conclusion

We have developed a reliable and consistent predictive model based on the identification of potential hub genes, which can be used to accurately forecast the prognosis of patients, thus providing direction for further clinical research and treatment.

Introduction

Pyroptosis is an inflammatory form of programmed cell death that is activated by the assembly of inflammasomes and regulated by inflammatory caspases (caspase-1/4/5/11). It is characterized by the rapid rupture of the plasma membrane, DNA fragmentation, and the release of proinflammatory intracellular contents (Valderrama et al., 2017; Fang et al., 2020). This process is mediated by the gasdermin superfamily, which includes GSDMA, GSDMB, GSDMC, GSDMD, and GSDME (or DFNA5) (Rogers et al., 2019). GSDME is suggested to have antitumor activity through phagocytosis and pyroptosis (Zhang et al., 2020). Pyroptosis has been reported to be involved in a variety of pathological conditions, such as inflammatory diseases and various types of cancer (Orning et al., 2018; Yu et al., 2019), and is thought to play a critical role in the initiation and progression of hepatocellular carcinoma (HCC) (Gaul et al., 2021; Huo et al., 2021). So far, only a few studies have been conducted to examine the impact of pyroptosis-related genes on the prognosis of hepatocellular carcinoma (HCC) (Chen et al., 2022).

Primary liver cancer is one of the most serious malignant tumors worldwide and the third leading cause of cancer-related deaths (Sung et al., 2021). Genome sequencing has uncovered the heterogeneity of both etiology and biology in primary liver cancer. This presents a daunting challenge for clinicians and researchers in providing precision treatment for this complex disease, which comprises multiple molecular subtypes. Hepatocellular carcinoma (HCC), the most common type of primary liver cancer, is typically linked to inflammation and makes up approximately 75–85% of cases, with intrahepatic cholangiocarcinoma accounting for the remaining 10–15% (Zheng et al., 2018). The development of HCC is a complex process involving chronic inflammation, liver injury, hepatocyte proliferation, and fibrosis (Krishna-Subramanian et al., 2019). Currently, the treatment of HCC is limited, and its prognosis is poor Curative ablative therapies are only suitable for patients with early-stage HCC. For advanced HCC, conventional chemotherapy and radiotherapy often suffer from drug resistance (Bollard et al., 2017; Ardelt et al., 2019).

In addition to tumor heterogeneity, the tumor microenvironment also plays a pivotal role in determining patient staging and assessing treatment outcomes (Foerster et al., 2022). The clinicopathological characteristics of HCC are the main prognostic factors. Owing to the complex heterogeneous character of HCC, a single biomarker combined with clinicopathological staging is not sufficient to predict the prognosis of patients with HCC. Although HCC is considered as a typical immunogenic cancer, immunotherapy has been shown to have limited effectiveness (Yu et al., 2019; Zhao et al., 2018). To date, there is no credible biomarker to predict the efficacy of immunotherapy for HCC, including programmed death-ligand 1(PD-L1), which has been found to be unable to predict the response to nivolumab and pembrolizumab (Pinter, Scheiner & Peck-Radosavljevic, 2021; Yau et al., 2022).

Therefore, an in-depth and systematic study of pyroptosis-related genes and their correlation with the occurrence and development of HCC may help to provide novel insights for specific diagnosis, clinical prevention, and effective therapy. It is particularly important to explore a combination of biomarkers and reliable prognostic models to optimize medical decision-making. There have been many bioinformatics studies on liver cancer (Gao et al., 2021). For example, Fu & Song (2021) obtained prognostic genes of HCC via analysis of the immune and biological functions of differential pyroptosis-related genes and verified them using qRT-PCR in vitro and in vivo; however, the sample size was small. Nevertheless, the study provided a new research idea connecting genes related to pyroptosis with those related to liver cancer. Zhang & Liu (2021) performed biomarkers using TCGA, Gene Expression Omnibus (GEO), and logistic regression analysis; their research has provided new signals for the diagnosis of liver cancer and prediction of survival rate of patients. This study also inspired us, Zhang & Liu (2021) used 1,374 differential methylation points, and we expanded the sample size to 8,958 DEGs and 197 pyroptosis-related genes. After comprehensive analysis of related differential genes, we will obtain more reliable gene analysis.

In this study, 37 differentially expressed genes related to pyroptosis were identified by taking the intersection of DEGs and pyroptosis-related genes, and their biological functions were explored. Subsequently, a risk prognostic model of HCC was constructed, and the biological function, drug sensitivity, and immune characteristics of DEGs were analyzed. Finally, we constructed a PPI network to analyze the key genes underlying the differences in groupings, and then conducted experimental validation to provide more therapeutic targets for clinical treatment of cancer.

Materials and Methods

Data collection

Gene expression profile data and clinical information for HCC were collected from TCGA (https://portal.gdc.cancer.gov/). The clinical data included age, sex, and survival status (Table S1). We obtained 369 tumor and 50 normal tissue samples. The copy number variation data for HCC were downloaded simultaneously. In addition, the Chip data set of GSE76426 (Grinchuk et al., 2018) from the GEO database was analyzed as a validation dataset and included 115 primary tumors (PTs) tissue samples and 53 adjacent non-tumor tissues (ANTTs) samples derived from 115 HCC patients, The dataset was obtained using the GPL 10558 Illumina Human HT-12 V4.0 expression beadchip sequencing platform. All the gene sets were derived from Homo sapiens. The overview of the workflow is shown in Fig. 1.

Figure 1 Work flowchart.

We took the intersection between 8,958 DEGs and 197 pyroptosis-related genes and got 37 pyroptosis-DEGs. The best prognostic gene was selected and the risk score was calculated. Then it is verified by GSE76427 and divided into low-risk group and high-risk group, 349 differentially expressed genes were identified from the two groups, the differential genes were functional annotation and immune analysis, finally the prognosis model was established.

Differentially expressed genes of HCC

To analyze the influence of gene expression values on the occurrence and development of HCC, differential gene analysis was performed on normal and tumor samples from TCGA dataset using the DESeq2 R package (Love, Huber & Anders, 2014). To define the threshold of DEGs, we applied the absolute value of log2Fold-change > 2 and Padj < 0.05; cut-off values of logFC > 2 and Padj < 0.05 and logFC < −2 and Padj < 0.05 were used for upregulated and downregulated genes, respectively. To uncover the influence of pyroptosis-related genes on biological functions, we extracted data from the GOBP_PYROPTOSIS dataset of the Genecards database (http://www.genecards.org) (Safran et al., 2010) and REACTOME_PYROPTOSIS dataset of the MSigDB (Liberzon et al., 2015) database. We then examined the intersection of pyroptosis-related genes and DEGs in HCC; the results were represented in the form of Venn diagrams. A volcano plot was constructed to show HCC-related and pyroptosis-related DEGs.

Construction of risk model for HCC

To analyze the effect of pyroptosis-related DEGs on the prognosis of patients with HCC, the expression and survival data of patients from TCGA database were analyzed. Prognostic risk genes for HCC were evaluated using univariate Cox proportional hazards regression analysis. After the prognostic risk genes for HCC were included in the model, we used the least absolute shrinkage and selection operator (LASSO) algorithm for dimension reduction and obtained prognostic feature genes. Each normalized gene expression value was weighted by penalty coefficients using LASSO-Cox analysis, and a risk score formula was established. Patients were then divided into high- and low-risk groups based on the mean risk score calculated as follows:

riskScore=∑iCoefficient (risk genei)*mRNA Expression (risk genei).

To verify the accuracy of the risk score, we used the GSE76427 dataset, which included patient gene expression and clinical data, to group patients and conduct a statistical analysis of the differences in survival.

Clinical prediction model based on risk model

We used a combination of the risk score combined and clinicopathological characteristics to assess the personalized prognosis of patients. Univariate and multivariate Cox regression analyses of risk score and clinicopathological characteristics were used to assess the predictive power of OS. Finally, a clinical predictive nomogram model was constructed by incorporating the risk score model and important clinicopathological parameters. To quantify the discriminative performance, we compared the nomogram-predicted probability with the observed actual survival rate and used a calibration curve to evaluate its nomogram performance.

Differentially expressed genes between the high-risk and low-risk groups

To analyze the effect of the risk score on HCC, DEG analysis of the samples that were divided into high- and low-risk groups from TCGA dataset was performed using the R package DESeq2 (Love, Huber & Anders, 2014). Subsequently, the significant differential genes were screened. The threshold values to determine the DEGs were as follows: the absolute values of the log2fold change (log2FC) > 2 and Padj < 0.05 were defined as upregulated DEGs and log2FC < −2 and Padj < 0.05 were defined as downregulated DEGs. The results were then represented as a volcano plot.

We obtained the dataset of HCC cell line-drug effects from the GDSC database (https://www.cancerrxgene.org/) (Yang et al., 2013). The gene expression data of high- and low-risk patients from TCGA database were subjected to drug sensitivity analysis using the R package OncoPredict (Maeser, Gruener & Huang, 2021). Subsequently, the sensitivity differences of the therapeutic drugs for HCC were compared between the high- and low-risk patients. In addition, to determine if the genes in the high- and low-risk patients with HCC showed copy number variation, GISTIC 2.0 in Genepattern (https://cloud.genepattern.org/) (Reich et al., 2006) was used to analyze the copy number variation of patients with HCC from TCGA database.

Assessment of biological characteristics

We used GO, KEGG (Ashburner et al., 2000; Kanehisa & Goto, 2000), GSEA, and GSVA (Hänzelmann, Castelo & Guinney, 2013) to analyze the biological characteristics of patients of different risk groups. GO annotation analysis and KEGG pathway enrichment analysis were performed on the DEGs using the clusterProfiler package (FDR < 0.05) (Yu et al., 2012). To investigate whether there were biological process differences between these two patient groups, we downloaded “c5.go.v7.4.entrez.gmt” and “c2.cp.kegg.v7.4.entrez.gmt” as reference gene sets from the Molecular Signatures Database (MSigDB, https://www.gsea-msigdb.org/gsea/msigdb) based on the gene expression profile data sets of patients with HCC. GSEA was performed, and the results were visualized using the clusterProfiler R package (P < 0.05). To study the variation in the biological processes of high-risk samples compared to that in those of low-risk samples, datasets of gene expression profiles based on TCGA were used to perform GSVA using the GSVA package in R. The reference gene set (“h.all.v7.4. symbols.gmt”) was downloaded from MSigDB. We then calculated an enrichment score for each sample in each pathway and examined the correlation between the enrichment score for patients and the risk score (P < 0.05).

Identification and correlation analysis of tumor-infiltrating immune cells

Analysis of immune cell infiltrates in tissues is a key component in the characterization of diseases and in the prediction of disease prognosis. The stomal and immune cell contents of high- and low-risk samples from TCGA dataset were estimated using the R package “estimate” (Yoshihara et al., 2013). The correlations between the risk score and ESTIMATE score were calculated. We evaluated the proportion of 22 immune cells in the immune microenvironment of high- and low-risk samples using the CIBERSORT algorithm (Newman et al., 2019). We set the number of permutations to 1,000 and used the Wilcoxon test to calculate the differences in proportions of immune cells between the high- and low-risk groups (P < 0.05).

Construction of protein-protein interaction (PPI) network

We used the STRING (Szklarczyk et al., 2019) dataset to construct PPIs of differentially expressed pyroptosis-related genes with a combined score of >400. Cytoscape (v3.7.0) was used for visualization of the PPI network, and Clue GO (Bindea et al., 2009) was used for functional annotation. Next, we used the cytoHubba plug-in to obtain hub genes (Chin et al., 2014) using the MCC method in the PPI network.

RNA extraction and real-time quantitative PCR assay

Total RNA was extracted from liver cancer and adjacent tissues (samples of liver cancer and adjacent tissues were collected from volunteers who underwent liver cancer surgery at the Second Affiliated Hospital of Dalian Medical University between January and December 2012, and each subject signed an informed consent form. All patient samples were approved by the Ethics Committee of the Second Affiliated Hospital of Dalian Medical University. No. 134.) using the M5 HiPer Universal RNA Mini Kit (Mei5 Biotechnology Co. Ltd, Beijing, China). We then used the PerfectStart® Uni RT&qPCR Kit (TransGen Biotech, Beijing, China) for reverse transcription into cDNA and performed qRT-PCR. The primers were as follows: (MDK) forward: 5′-GCAACTGGAAGAAGGAGT-3′, and MDK reverse: 5′-TGGCACTGAGCATTGTAG-3′.

Immunohistochemistry

Immunohistochemistry was performed on liver cancer and adjacent tissues (the source of sample is the same as that of qRT-PCR experiment). We performed dewaxing with xylene, followed by a gradient dehydration with alcohol, and then inactivated the endogenous peroxidase with hydrogen peroxide. The sections were then blocked and treated with primary antibodies against MDK (anti-Midkine; Abcam, Cambridge, UK). The antibody was diluted with PBS at a ratio of 1:600, and then incubated for 12 h at 4 °C. To each film, 100 µL of working fluid (PV-9000; Beijing Zhongshan Golgen Bridge Biological Technology Co. Ltd, Beijing, China) was added and then incubated at 37 °C for 20 min. The solution was dehydrated again after dyeing, and the film was sealed.

Statistic analysis

All data were processed and analyzed using R software (version 4.1.1). Continuous variables were compared between the two groups. For normally distributed variables, we used independent t-tests; for non-normally distributed variables, we used the Mann-Whitney rank-sum test (Li et al., 2022). Correlations between different genes were analyzed using the Pearson correlation coefficient calculation. Survival analysis was performed using the R survival package. Survival differences are shown using Kaplan–Meier survival curves. The significance of the difference in survival between the two groups was assessed using the log-rank test. Univariate and multivariate Cox regression analyses were used to identify independent prognostic factors. All statistical P-values were two-sided, and P < 0.05 was considered statistically significant.

Results

Differentially expressed pyroptosis-regulated factors and the effect on their biological processes

To reveal the biological differences between HCC and normal samples from transcriptomes, we performed differential gene expression analysis in HCC and normal samples. After screening, 8,958 genes were defined as significant DEGs, comprising 7,030 upregulated and 1,928 downregulated genes (Fig. 2A). In total, 197 pyroptosis-related genes were obtained from the GeneCards database. The intersection of DEGs and pyroptosis-related genes revealed 37 differentially expressed pyroptosis-related genes (22 upregulated and 16 downregulated; Figs. 2B, 2C).

Figure 2 Differentially expressed genes and their biological functions.

(A and B) The volcanic map shows differentially expressed genes and differentially expressed pyroptosis-related genes. The abscissa is log2FoldChange and the ordinate is −log10 (adjusted P-value). The up-regulated differentially expressed genes are represented by red nodes, the down-regulated differentially expressed genes are represented by blue nodes, and the genes with insignificant differentially expressed genes are represented by black nodes. (C) A Venn diagram of differentially expressed genes and pyroptosis-related genes. The blue circle indicates the differentially expressed genes in the TCGA data set, and the orange circle indicates the pyroptosis gene. (D) The GO of differentially expressed pyroptosis-related genes. The color of the bar graph indicates the zscore of GO term, and the size of z-score indicates the activation or suppression of GO term. (E–G) The results of BP, CC and MF in the GO of differentially expressed pyroptosis-related genes. (H) The results of KEGG pathway enrichment analysis of differentially expressed pyroptosis-related genes.

Functional enrichment was performed for differentially expressed pyroptosis-related genes (Fig. 2D). The results showed that the differential genes were related to biological processes (Fig. 2E) including digestion, maintenance of gastrointestinal epithelium, digestive system process, epithelial structure maintenance, and O-glycan processing; cellular components (Fig. 2F) including extracellular space, extracellular region, endosome membrane, and plasma membrane; and molecular functions including interleukin-1 receptor binding, lipoteichoic acid binding, G-protein coupled adenosine receptor activity, lipopolysaccharide binding, and identical protein binding (Fig. 2G). Furthermore, KEGG pathway enrichment analysis indicated that the differentially expressed pyroptosis-related genes were significantly enriched in the pertussis, Salmonella infection, toll-like receptor signaling, and tuberculosis pathways (Fig. 2H).

Construction of risk models and prognostic analysis

To analyze the effects of differentially expressed pyroptosis-related genes on the prognosis of patients with HCC, we used univariate Cox regression analysis and identified 45 prognostic risk genes of HCC. The prognosis-associated risk genes of HCC were included in the LASSO-Cox analysis to choose 12 genes with the best prognostic value. We subsequently performed a correlation analysis of the expression of the 12 characteristic genes, and the results showed a high level of correlation between these genes (Fig. 3A). The penalty coefficients of characteristic genes were calculated based on LASSO-Cox analysis (ADRA1A: −0.023, GPRC6A: 0.037, MAGEA8: 0.034, MDK: 0.024, MMP9: 0.019, NETO2: 0.027, OLFM3: 0.031, PYCR1: 0.020, RET: 0.029, and TNFRSF4: 0.065). The risk score was calculated by multiplying the expression level of a gene with its corresponding coefficient and then adding them together. Patients with HCC were then divided into high- and low-risk groups based on their mean risk scores. In addition, principal component analysis revealed that the 12 characteristic genes could better distinguish patients in both TCGA and GSE76427 datasets (Figs. 3B, 3C). Kaplan–Meier analysis revealed worse OS scores in patients with high risk scores (log-rank P < 0.0001, Fig. 3D). The risk score of each patient with HCC in GSE76427 was then computed, and the Kaplan–Meier analysis revealed that patients with a high-risk score showed worse OS (log-rank P < 0.05, Fig. 3E).

Figure 3 Construction of risk scoring model.

Construction of risk scoring model. (A) This figure shows a correlation analysis of characteristic genes in HCC. (B and C) The photos represent the PCA analysis of characteristic genes in both TCGA-HCC and GSE76427 datasets. Red indicates low-risk groups and blue indicates high-risk groups. (D and E) Kaplan–Meier analysis revealed the effect of risk score on the overall survival of patients with HCC in TCGA-HCC and GSE76427. Red indicates the low-risk group and blue indicates the high-risk group.

Construction of clinical prediction model based on risk score

We evaluated the prognostic impact of the risk score in patients with HCC. Univariate and multivariate Cox analyses revealed that the risk score was an independent prognostic factor for patients with HCC (Figs. 4A, 4B). Different clinicopathological features were combined into a risk score to generate a prediction model nomogram for predicting the OS of patients with HCC (Fig. 4C). The calibration curve showed a good prediction of 1-, 2-, and 3-year survival for patients using the model (Figs. 4D–4F). Concurrently, the results of time-dependent receiver operating characteristic curve analysis showed that the predicted percentages of 1-, 3-, and 5-year survival were 77.8%, 79.8%, and 82.4%, respectively (Fig. 4G). The decision curve analysis supported the model, which could offer greater prognostic gains for patients (Fig. 4H).

Figure 4 Analysis of the predictive ability of risk score on the prognosis of HCC patients.

(A) Univariate Cox analysis. (B) Multivariate Cox analysis. (C) Forecast model nomogram. (D–F) The calibration curve of the nomogram of the prediction model. The abscissa is the survival predicted by the nomogram, and the ordinate is the actual observed survival. The curve shows the prediction of the prognosis of the model for HCC patients for 1, 3 and 5 years. (G) Time-ROC curve for predicting 1-, 3- and 5-year survival of patients with hepatocellular carcinoma by nomograph model, (H) DCA curve for 1-, 3- and 5-year survival of patients with HCC by nomograph model.

Divergence analysis between high-risk group and low-risk group

Owing to the clear distinction in survival between the high- and low-risk groups, we performed differential expression analysis of the expressed genes in the high- and low-risk groups. In total, 349 DEGs were identified. Of these, 51 genes were upregulated and 298 were downregulated (Fig. 5A). Concurrently, the DEGs were subjected to dimensionality reduction using principal component analysis. The results showed a significant difference between the high- and low-risk groups (Fig. 5B). We then analyzed the effect of differentially expressed mRNA on the biological functions associated with the high- and low-risk groups. GO biological function annotation analysis of the DEGs showed that these genes were mainly enriched in digestion, maintenance of gastrointestinal epithelium, digestive system process, epithelial structure maintenance, O-glycan processing, and other biological processes (Fig. 5C); catenin complex, GABAergic synapse, Golgi lumen, dendrite membrane, apical part of cell, and other cellular components (Fig. 5D); and ligand-gated anion channel activity, chloride channel activity, hormone activity, anion channel activity, and other molecular functions (Fig. 5E; Table S2). Additionally, KEGG pathway enrichment analysis of the DEGs showed that these genes were mainly enriched in pathways associated with neuroactive ligand-receptor interaction, taste transduction, pancreatic secretion, taurine and hypotaurine metabolism, and the cAMP signaling pathway (Fig. 5F; Table S3). We also analyzed the copy number differences between the high- and low-risk groups. The results revealed that the copy number amplification of patients assigned to the high-risk group was significantly lower than that of those assigned to the low-risk group, and the high-risk group included more chromosomal deletions (Fig. 6).

Figure 5 Differentially expressed genes between high-risk group and low-risk group.

(A) The figure shows the volcanic map of differentially expressed genes between high-risk group and low-risk group. The abscissa is log2FoldChange and the ordinate is −log10 (adjusted P-value). The red node indicates the up-regulated differentially expressed genes, the blue node indicates the down-regulated differentially expressed genes, and the black node indicates the genes that are not significantly differentially expressed. (B) The PCA analysis of differentially expressed genes between high-risk group and low-risk group. (C–E) The results of BP, CC and MF in the GO biological function annotation analysis of the differentially expressed genes. (F) The result of KEGG pathway enrichment analysis of differentially expressed genes.

Figure 6 Copy number differences between high-risk and low-risk groups.

(A–D) Genes with significant amplification and deletion. Error detection rate (Q value) and GISTIC2.0 of the change score (x-axis) corresponds to the genomic position (y-axis). The dotted line indicates the centromere. The green line indicates the 0.25 Q cut-off point for determining significance. These figures represent the copy number amplification of patients in the high-risk group, copy number deletion in high-risk group, copy number amplification in low-risk group and copy number deletion in patients in the low-risk group.

To analyze the influence of risk score on the treatment of HCC, we integrated the gene expression data of high- and low-risk patients in TCGA dataset with drug sensitivity for HCC from the GDSC database. We identified drugs that were more sensitive in high- and low-risk patients and had lower IC50 values. Among the drugs currently used for the treatment of HCC, IGFR_3801, tanespimycin, docetax, and lestaurtinib were more sensitive in high-risk patients (Fig. 7A), whereas belinostat, ispinesib mesylate, shikonin, and midostaurin were more sensitive in low-risk patients (Fig. 7B). Among the currently used drugs for other cancers, trametinib, mitomycin-C, rapamycin and obatoclax mesylate were more sensitive in high-risk patients (Fig. 7C), whereas pevonedistat_1529, methotrexate_1008, foretinib_308, and AZD8055_1059 were more sensitive in low-risk patients (Fig. 7D).

Figure 7 Drug sensitivity analysis of patients in high-risk and low-risk groups.

(A–D) The horizontal axis represents the patient grouping, and the vertical axis represents the log10 (IC50) of the drug. These pictures in turn show the more sensitive hepatocellular carcinoma drugs in the high-risk group, the more sensitive hepatocellular carcinoma drugs in the low-risk group, the more sensitive other cancer drugs in the high-risk group and the more sensitive other cancer drugs in the low-risk group.

Next, we performed GSEA of the gene expression data of the high- and low-risk groups. We found that the patients in the high- and low-risk groups showed significant differences mainly in the following biological processes: inhibitory extracellular ligand-gated ion channel activity, limb bud formation, condensed chromosome outer kinetochore, spindle elongation, regulation of chloride transport, and other processes were inhibited, whereas very low-density lipoprotein particle remodeling, triglyceride-rich lipoprotein particle remodeling, complement activation lectin pathway, platelet dense granule lumen, alcohol dehydrogenase were activated (Figs. 8A and 8B). Simultaneously, we found that primary bile acid biosynthesis, fatty acid metabolism, complement and coagulation cascades, glycine serine and threonine metabolism, renin angiotensin system, and other pathways were activated, whereas some pathways, such as cell cycle, ECM receptor interaction, neuroactive ligand receptor interaction, gap junction, and axon guidance, remained inactivated (Figs. 8C and 8D; Tables S4 and S5).

Figure 8 GSEA analysis of gene expression data of high- and low-risk groups.

(A) The GSEA-GO analysis of TCGA-HCC dataset. The abscissa is gene ratio, the ordinate is GO terms, and the color represents-log10 (p value). (B) The GSEA-KEGG analysis of TCGA-HCC dataset. The abscissa is gene ratio, the ordinate is GO terms, the node size represents the number of genes enriched in GO terms, and the node color represents −log10 (p value). (C) The first three items of GSEA-GO analysis of TCGA-HCC dataset. (D) The first three items of TCGA-HCC analysis of TCGA-HCC dataset. (E) The heat map shows the GSVA analysis of high-risk and low-risk groups. The horizontal axis is the patient ID and the vertical axis is the hallmark gene set.

We further used GSVA to calculate the DEGs of high- and low-risk patients from TCGA database to determine their effect on biological characteristics and oncogenic signaling pathways. The results revealed that the risk score was closely correlated with the activity of multiple oncogenic signaling pathways (Fig. 8E). Among them, there was a significant negative correlation between risk score and hallmark KRAS signaling DN, hallmark pancreas beta cells, hallmark coagulation, hallmark xenobiotic metabolism, and hallmark bile acid metabolism, whereas there was a significant positive correlation between risk score and hallmark G2M checkpoint, hallmark E2F targets, hallmark MYC targets V2, hallmark DNA repair, and hallmark MYC targets V1.

Immune characterize differences in high-risk and low-risk groups

We analyzed the differences in immune cell content in the high- and low-risk groups (Fig. 9A), and the high- and low-risk groups showed significant differences in multiple immune cells (Fig. 9B). The risk score was positively correlated with B cells memory, dendritic cells resting, macrophages M0, and regulatory T cells (Tregs), and negatively correlated with resting mast cells, monocytes, and T cells gamma delta (P < 0.05, Fig. 9C). Additionally, we calculated the correlation between immune cells and patients in the high- and low-risk groups. We found that there was a positive correlation between T cells CD4 memory resting/dendritic cells resting in patients in the high-risk group (Fig. 9D), while there was a negative correlation between B cells memory, dendritic cells resting, neutrophils, Tregs, macrophages M0 in patients in the low-risk group (Fig. 9E).

Figure 9 Analysis of immune cell infiltration in high-risk and low-risk groups.

(A) The heat map shows the distribution of immune cells in different risk groups. The horizontal axis is the patient ID and the vertical axis is the proportion of immune cells. (B) The results showed the difference of immune cell content among different risk groups. The horizontal axis is immune cells, the vertical axis is immune cell content, blue indicates patients in low-risk group, and orange indicates patients in high-risk group. (C) The results showed the correlation analysis between the prognostic risk score and the content of immune cells. The horizontal axis is the immune cells significantly related to the prognostic risk score, the vertical axis is the correlation score, and the color of the bar graph indicated the significance of the correlation, P < 0.05 means statistically significant. (D and E) This is the heat map of immune cell correlation between high-risk and low-risk groups. Blue indicates positive correlation and red indicates negative correlation.

We also analyzed the correlations between the risk score of HCC and the immune score, stromal score, tumor purity, and ESTIMATE score, among which there was a significant positive correlation between the risk score and stromal score (P < 0.05, Fig. 10A). We constructed a PPI network of DEGs in patients in the high- and low-risk groups (Fig. 10B). Visualization of the PPI network was carried out with Cytoscape, which included 159 DEGs and 272 PPI pairs in total. The top five genes that interacted closely with other DEGs were MUC5AC (interacted with 16 DEGs), MUC5B (interacted with 16 DEGs), MUC1 (interacted with 15 DEGs), CFTR (interacted with 14 DEGs), and CHGA (interacted with 13 DEGs). In addition, DEGs related to the PPI network were mainly enriched in functions related to calcium-ion regulated exocytosis, digestive system processes, and cochlear development (Fig. 10C). The hub genes of the PPI network were screened using the cytoHubba plugin, and these genes are more likely to be the key genes causing differences between the high- and low-risk groups (Fig. 10D).

Figure 10 PPI network.

(A) Violin chart shows the difference of stromal score between high-risk and low-risk groups. Blue indicates patients in low-risk group, orange indicates patients in high-risk group, P < 0.05 means statistically significant. (B) A protein-protein interaction network of differentially expressed genes in patients of high- and low-risk groups. (C) The results of clueGO enrichment analysis in the protein-protein interaction network related to differentially expressed genes. (D) The hub genes in the protein-protein interaction network related to differentially expressed genes.

High expression of MDK in hepatocellular carcinoma

From the RT-qPCR results (Fig. 11), it was clear that the expression of MDK in HCC is much higher than that in adjacent tissues. The immunohistochemical analysis (Fig. 12) also showed that MDK was highly expressed in HCC tissues compared to that in normal liver tissues.

Figure 11 qRT-PCR results.

Detection of MDK expression in hepatocellular carcinoma and adjacent tissues by qRT-PCR. An asterisk (*) indicates P < 0.05 vs paracancerous, n = 5.

Figure 12 Immunohistochemical results.

The three images on the left are cancer tissue after immunohistochemical staining with MDK antibody, and the three images on the right are cancer tissue after immunohistochemical staining with MDK antibody.

Discussion

Recently, there has been an upsurge in the number of individuals diagnosed with liver cancer (Foerster et al., 2022; Rumgay et al., 2022). HCC has become one of the major leading causes of cancer-related mortality in recent years, particularly in China, where the 5-year survival rate of HCC patients remains distressingly low, estimated at less than 13% (Wang & Wei, 2020). The pathogenesis of HCC and its specific underlying mechanisms remain under investigation (Ye et al., 2021). The diagnosis and prognosis of patients with HCC is largely based on the Barcelona Clinic Liver Cancer staging the aspects of pathology and pathology (Bruix, Reig & Sherman, 2016). In addition, alpha-fetoprotein (AFP) levels are used to identify HCC in its early stages. Its specificity can reach 80–94% in chronic viral hepatitis and cirrhosis, while its sensitivity is less than 70% in both chronic viral hepatitis and cirrhosis (Wang & Wei, 2020; Daniele et al., 2004; Galle et al., 2019). Accurately predicting the OS of patients with HCC is important for clinical decision-making. However, there are currently no effective and reliable prognostic biomarkers for patients with HCC (Wu et al., 2021).

We obtained the genes with the best prognostic value using single-factor Cox regression analysis and LASSO Cox analysis. Afterwards, we conducted a more detailed study on the MDK gene. MDK, also known as neurite growth-promoting factor 2 (NEGF2), is a small heparin-binding growth factor with a molecular weight of approximately 13 kDa. The human MDK gene is located on chromosome llp.11.2 (57) (Zhu et al., 2013; Gowhari Shabgah et al., 2021). MDK has been found to play an important role in nervous system development during embryonic development, as well as in the regulation of inflammatory reactions associated with autoimmune diseases and cancer. An increasing number of studies have sdemonstrated that MDK is a multifunctional molecule involved in the progression of various tumors, including proliferation, promotion of angiogenesis, and suppression of apoptosis (Du et al., 2016; Kato et al., 2000; Muramatsu, 2010). It has been reported that MDK also plays a role in the occurrence, development, metastasis, and prognosis of liver cancer (Muramatsu, 2010). Further more, some studies have found that the sensitivity of MDK for the diagnosis of HCC could reach 85% through meta-analysis (Zhang et al., 2019), and other studies have shown that the sensitivity can reach more than 80% in the early stages of liver cancer and AFP-negative liver cancer (Luo et al., 2020). In addition, many liver cancers are associated with HBV and HCV infections, and it has been suggested that MDK may have an impact on hepatitis-associated HCC (Lu et al., 2020). We demonstrated through qRT-PCR and immunohistochemistry that MDK is highly expressed in HCC cells, which suggests that MDK has a potential to become a new therapeutic target in the near future, providing a new method of clinical treatment. In addition to its role in early diagnosis, high MDK expression can indicate poor prognosis and the possibility of recurrence (Hodeib et al., 2017). Many studies have shown that MDK is a biomarker of liver tumors, which is consistent with our findings. MDK can be utilized as a novel biomarker for hepatocellular carcinoma (HCC) to enhance early diagnosis accuracy and as a potential target for clinical therapy (Tsuchiya et al., 2015; Darmadi, Ruslie & Pakpahan, 2022; Zhang et al., 2020).

Through this study, we found that the OS of the high-risk group was relatively worse than that of the low-risk group. To further explore the differences between the high- and low-risk groups, we analyzed the DEGs between these groups. We obtained hub genes by constructing PPIs of differentially expressed pyroptosis-related genes. These hub genes play a key role in the risk grouping of liver cancer. The hub genes were SST, KRT19, CRHR1, GRM5, KRT20, AGR2, MUC6, TFF1, TFF2, CLCA1, CFTR, CHGA, MUC5A, TAC1, MUC1, MUC5B, GALNT12, FOXJ1, and SCGB1A1 (Fig. 10D). In the PPI of differentially expressed pyroptosis-related genes, we found that MUC5AC interacted the most with other DEGs. MUC5AC is a mucin, which is a product of secretory cells. MUC5AC is located on the chromosome 11 (11p15) (Van Seuningen et al., 2001). It is the main component of airway mucus, is also expressed in intestinal epithelial cells, and plays a protective role. In recent years, it has been found to be abnormally expressed in various diseases, such as asthma and malignant mucinous lung tumor cells. It can also be used as an indicator of the prognosis of colorectal cancer (Bonser & Erle, 2017; Lin et al., 2021; Hazgui et al., 2021). In the past, researchers thought that HCC does not produce mucin (Vocka et al., 2015). However, recent studies have shown that HCC is capable of producing MUC5AC (Wang et al., 2017; Xuan et al., 2016; Pabalan et al., 2019). Our study provides evidence of the potential involvement of MUC5AC in the etiology of hepatocellular carcinoma (HCC). Further exploration of its possible mechanisms is warranted to identify prospective therapeutic targets or prognostic markers.

MUC1 is also a widely recognized oncogene. MUC1 is a one-way type I transmembrane protein that is mostly expressed in epithelial cells such as those in the stomach and colorectal and respiratory tracts. It has the same protective role as MUC5AC (Ochiai et al., 2020). MUC1 is an aberrantly glycosylated and overexpressed protein in HCC which is found to have critical roles in various epithelial cancers, distinct functions in both normal and abnormal cells, and is overexpressed in the case of HCC (Nath & Mukherjee, 2014). Its mechanism is to promote the occurrence and development of tumors via the JNK/TGF signaling pathway (Wang et al., 2017). In this study, through PPI analysis, MUC1 is the third hub genes that interacted closely with other DEGs in HCC. Therefore, MUC1 may be a biological marker to diagnose and judge the prognosis of HCC, and may be a therapeutic target for HCC in the future.

By analyzing the biological characteristics of patients in different risk groups, we found that the DEGs related to the PPI network were mainly enriched in biological processes such as digestive system processes, epithelial structure maintenance, and O-glycan processing. The hub genes of TFF1 and TFF2 obtained by us are also mainly expressed in the digestive system. The TFF family gene is located on the chromosomes 21q22.3. The TFF family includes TFF1, TFF2, and TFF3 (Chinery, Williamson & Poulsom, 1996). TFF1 is mainly expressed in digestive tract epithelial cells (Zhang et al., 2019), and is mainly involved in the protection of the gastrointestinal tract via the repair of epithelial cells; it is highly expressed in gastric and colorectal cancer (Vocka et al., 2015). Yosuke et al. found that TFF1 has a tumor inhibitory effect. It could inhibit the proliferation of liver cancer cells and induce apoptosis and inhibit the cell cycle by negatively regulating the β-catenin signaling pathway (Ochiai et al., 2020; Hoffmann, 2009). TFF2 is mainly expressed in the mucous cells of the digestive tract and plays a protective role in promoting intestinal epithelial cell migration (Ge et al., 2020). TFF2 is expressed at low levels in gastric cancer and is related to methylation. Moreover, TFF2 plays an inhibitory role in breast and pancreatic cancer (Ge et al., 2020; Yamaguchi et al., 2016; Ishibashi et al., 2017). At present, research into the role of TFF1 and TFF2 in the progression of HCC is limited, and further studies are needed to ascertain the exact mechanism of action.

This study had some limitations. First, our data were collected from TCGA, which lacks relevant clinical data and its analysis. Moreover, we applied multiple datasets in this study, and there may have been batch differences. Therefore, the statistical power may be low. In order to compensate for this limitation, we conducted external dataset validation, combined with our hospital samples for joint analysis, and then validated the results with further molecular experiment. Second, based on bioinformatics analysis, we identified the hub genes that are involved in inducing the difference between high- and low-risk groups of patients with HCC; However, to gain a molecular and tissue-level understanding of the roles of the hub genes, we still need to conduct experiments such as Western Blotting, a Cell Scratch Assay, and a Cell Migration Experiment.

This study established a risk prediction model by exploring the role of differentially expressed pyroptosis-related genes on the prognosis of HCC. By combining clinical information with the biological and immune characteristics and protein-protein interaction (PPI) of the DEGs, we conducted a comprehensive analysis to gain insight into the pathogenesis of HCC This study enhances our understanding of the molecular mechanisms of HCC, and the hub genes are potential therapeutic targets for HCC and offer novel concepts for clinical treatment. Nonetheless, the precise pathogenesis and molecular targets of HCC need to be substantiated by future studies.

Conclusions

In this study, an integrated bioinformatics analysis was conducted to explore the potential influence of pyroptosis-related DEGs on HCC. Subsequently, a reliable prognostic model incorporating the risk score and clinicopathological features was constructed, which was able to effectively predict and assess the 1-, 3- and 5-year survival rates of HCC patients. Moreover, experimental validation was conducted to further confirm the higher expression of MDK, a key gene in HCC, thus providing a potential biomarker for diagnosis and a therapeutic target for clinical research and therapy.

Supplemental Information

Supplemental Information 1 TCGA-HCC patient baseline table.

Click here for additional data file.

Supplemental Information 2 Go enrichment analysis of differentially expressed genes between high-risk and low-risk groups.

Click here for additional data file.

Supplemental Information 3 KEGG enrichment analysis of DEGs of differentially expressed genes between high-risk and low-risk groups.

Click here for additional data file.

Supplemental Information 4 GSEA analysis between high-risk and low-risk groups (inhibited biological function).

Click here for additional data file.

Supplemental Information 5 GSEA analysis between high-risk and low-risk groups (promoted biological functions).

Click here for additional data file.

Supplemental Information 6 The Expression Level of MDK Gene Measured by Quantitative PCR.

Click here for additional data file.

Supplemental Information 7 Test Plan.

Click here for additional data file.

Abbreviations

HCC Hepatocellular carcinoma

OS overall survival

GO Gene Ontology

KEGG Kyoto Encyclopedia of Genes and Genomes

GSEA Gene Set Enrichment Analysis

GSVA Gene Set Variation Analysis

PPI protein-protein interaction

qRT-PCR real-time quantitative PCR

HBV hepatitis B virus

PRG differential pyroptosis related genes

TCGA The Cancer Genome Atlas

GEO Gene Expression Omnibus

DEGs differentially expressed genes

LASSO least absolute shrinkage and selection operator

OS overall survival

MSigDB Molecular Signatures Database

MDK Midkine

Additional Information and Declarations

Competing Interests

Author Contributions

Human Ethics

Data Availability

The authors declare that they have no competing interests.

Bingbing Shang performed the experiments, analyzed the data, prepared figures and/or tables, authored or reviewed drafts of the article, and approved the final draft.

Ruohan Wang performed the experiments, analyzed the data, prepared figures and/or tables, authored or reviewed drafts of the article, and approved the final draft.

Haiyan Qiao performed the experiments, analyzed the data, prepared figures and/or tables, authored or reviewed drafts of the article, and approved the final draft.

Xixi Zhao performed the experiments, authored or reviewed drafts of the article, and approved the final draft.

Liang Wang conceived and designed the experiments, analyzed the data, prepared figures and/or tables, authored or reviewed drafts of the article, and approved the final draft.

Shaoguang Sui conceived and designed the experiments, analyzed the data, prepared figures and/or tables, authored or reviewed drafts of the article, and approved the final draft.

The following information was supplied relating to ethical approvals (i.e., approving body and any reference numbers):

Ethics Committee of the Second Affiliated Hospital of Dalian Medical University.

The following information was supplied regarding data availability:

The Expression Level of MDK Gene Measured by Quantitative PCR are available in the Supplemental File.

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
