# Peer review of "Multi-omics analysis of pyroptosis regulation patterns and characterization of tumor microenvironment in patients with hepatocellular carcinoma"

_PeerJ, doi:10.7717/peerj.15340_

## Round 0.1 · original submission · Major Revisions

Please see the comments and feedback from both reviewers. Both reviewers appreciated the breadth of the study. However, to present this work as a comprehensive, in-depth study, authors need to redo figures, rewrite most of the sections in this manuscript, and clearly explain limitations and what authors have done to address those limitations to maintain the scientific validity of findings. Therefore, I am recommending Major Revisions. Please make sure to address all the comments by reviewers, and provide a point-by-point response.

·

Basic reporting

I would highly recommend to reorganize and present the manuscript in fluent and succinct English.
Literature references are cited well however, at some places a few relevant references are required.
I would recommend the authors to make figures to contain more information about the data that is represented and reorganize the figures to flow with the text.
It would be helpful to state a clear cut hypothesis or a trajectory of the study and how it approached the core concept in abstract and introduction.
In my opinion, a good round of rewriting would help increase the weightage of the study.

Experimental design

Experimental approach is well thought but not represented accordingly. It might be better to put the study findings in the broader context of the field. Methods section is quite comprehensive.

Validity of the findings

As pointed by the authors in the discussion, collected data from TCGA and other datasets had significant limitations with respect to batch differences and relevant depth of information available.
It would be important that it is highlighted how this limitation was overcome or accounted for in this study.

Additional comments

NO.

Reviewer 2 ·

Basic reporting

.

Experimental design

.

Validity of the findings

.

Additional comments

This is a comprehensive study. However, the authors failed to describe and tell the story.

Abstract :
In the first line, the usage of 'Dangerous' should be avoided. Carcinoma itself is self-explanatory; please rewrite the sentence.
The Results section in the abstract needs extensive revision.

- Hepatocellular carcinoma (HCC), Liver Hepatocellular Carcinoma (LIHC) can be used interchangeably; authors should be consistent while using these
- Authors should provide keywords or link to TCGS and how they obtain 369 tumors and 52 controls. The data downloaded section should be Revised and elaborated. The name of the heading should be 'Data Collection' instead of 'Data downloaded"'
- Figure 1 is not mentioned in the text but in the conclusion.
- links should be provided in the text
- Why LASSO is preferred by authors compared to other methods
- Short form should be expanded when used for the first time in the main body of text.

- In section 2.9, the information about human samples is unclear and should be elaborated.
- Section 2.10 is missing.
- Section 2.11 should be revised after describing what samples they are talking about. What tissue and number.

- Figure 2C, please do confirm that the Venn diagram circle names are correct.
- Pair plot can be removed from the 3a, and authors can add a correlation matrix.
- Description of Figure 4G-H comes before Figure 4A, similar to figure 5B comes before 5A.
- Authors did not clearly mention and justify using the validation dataset. Needs extensive revision.

---

## Round 0.2 · accepted · Accept

I am writing to inform you that your manuscript - Multi-omics Analysis of Pyroptosis Regulation Patterns and Characterization of Tumor Microenvironment in Patients with Hepatocellular Carcinoma - has been Accepted for publication. Congratulations!

Reviewer 2 ·

Basic reporting

The revised manuscript has improved a lot and authors have addressed the concerns I had, however before final publication, I would ask to re-revise the manuscript for English language.

Experimental design

No comments

Validity of the findings

No Comments

Additional comments

No comments